# Subacute Ingestion of Caffeine and Oolong Tea Increases Fat Oxidation without Affecting Energy Expenditure and Sleep Architecture: A Randomized, Placebo-Controlled, Double-Blinded Cross-Over Trial

**DOI:** 10.3390/nu12123671

**Published:** 2020-11-28

**Authors:** Simeng Zhang, Jiro Takano, Norihito Murayama, Morie Tominaga, Takashi Abe, Insung Park, Jaehoon Seol, Asuka Ishihara, Yoshiaki Tanaka, Katsuhiko Yajima, Yoko Suzuki, Chihiro Suzuki, Shoji Fukusumi, Masashi Yanagisawa, Toshio Kokubo, Kumpei Tokuyama

**Affiliations:** 1International Institute for Integrative Sleep Medicine (WPI-IIIS), University of Tsukuba, Tsukuba, Ibaraki 305-8575, Japan; evergreenmeng@gmail.com (S.Z.); tominaga.morie.ft@un.tsukuba.ac.jp (M.T.); abe.takashi.gp@u.tsukuba.ac.jp (T.A.); park.insung.ge@u.tsukuba.ac.jp (I.P.); asuka.ishihara@gmail.com (A.I.); yoshi.tanaka.5212@gmail.com (Y.T.); yoko.12.suzu@gmail.com (Y.S.); s1821271@gmail.com (C.S.); fukusumi.shoji.fw@u.tsukuba.ac.jp (S.F.); yanagisawa.masa.fu@u.tsukuba.ac.jp (M.Y.); kokubo.toshio.fp@u.tsukuba.ac.jp (T.K.); 2Research Institute, Suntory Global Innovation Center Ltd., Soraku, Kyoto 619-0284, Japan; Jiro_Takano@suntory.co.jp (J.T.); Norihito_Murayama@suntory.co.jp (N.M.); 3R&D Center for Tailor-Made QOL, University of Tsukuba, Tsukuba, Ibaraki 305-8550, Japan; tsukubaseol@gmail.com; 4Faculty of Pharmaceutical Sciences, Josai University, Saitama 350-0295, Japan; yajimamarch@yahoo.co.jp

**Keywords:** whole room indirect calorimetry, respiratory quotient, fat oxidation, carbohydrate oxidation, body temperature, caffeine tolerance

## Abstract

Ingesting oolong tea or caffeine acutely increases energy expenditure, and oolong tea, but not caffeine, stimulates fat oxidation. The acute effects of caffeine, such as increased heart rate and interference with sleep, diminish over 1–4 days, known as caffeine tolerance. During each 14-day session of the present study, 12 non-obese males consumed oolong tea (100 mg caffeine, 21.4 mg gallic acid, 97 mg catechins and 125 mg polymerized polyphenol), caffeine (100 mg), or placebo at breakfast and lunch. On day 14 of each session, 24-h indirect calorimetry and polysomnographic sleep recording were performed. Caffeine and oolong tea increased fat oxidation by ~20% without affecting energy expenditure over 24-h. The decrease in the respiratory quotient by oolong tea was greater than that by caffeine during sleep. The effect of oolong tea on fat oxidation was salient in the post-absorptive state. These findings suggest a role of unidentified ingredients in oolong tea to stimulate fat oxidation, and this effect is partially suppressed in a postprandial state. Two weeks of caffeine or oolong tea ingestion increased fat oxidation without interfering with sleep. The effects of subacute ingestion of caffeine and oolong tea differed from the acute effects, which is a particularly important consideration regarding habitual tea consumption.

## 1. Introduction

The various types of tea produced from a single plant species, Camellia sinensis, are distinguished by the processing technique: unoxidized tea as green tea, half-oxidized tea as oolong tea, and fully oxidized tea as black tea. In the course of oxidation catalyzed by endogenous enzymes, catechins are transformed to polymerized polyphenols, and caffeine is slightly decomposed [1], alterations that might yield benefits and/or risks on diverse physiologic functions.

The effects of tea on energy metabolism have been assessed in parallel with the effects of caffeine. Many studies report that accumulated energy expenditure over 24 h is increased by caffeine ingestion [2,3,4], although one study found no effect [5]. The effect of caffeine ingestion on accumulated fat oxidation over 24 h, however, was not examined in any of these four studies [2,3,4,5]. Ingestion of green tea extract increases 24-h energy expenditure and decreases the 24-h respiratory quotient (RQ) [5]. Another study found that 3 days of oolong tea ingestion increased the accumulated energy expenditure and fat oxidation over 24 h, whereas caffeine alone did not significantly increase 24-h fat oxidation [4]. Thus, unidentified ingredient(s) of tea other than caffeine affects energy metabolism, particularly fat oxidation. Importantly, in these previous studies on energy metabolism, caffeine and/or tea were ingested for only 1 [2,3,5] or 3 days [4]. Known as caffeine tolerance, the acute effect of caffeine to increase blood pressure and heart rate diminishes over 1–4 days when caffeine ingestion is continued [6]. Together, these findings warrant further investigation of the subacute and chronic effects of tea and caffeine consumption on energy metabolism.

Caffeine inhibits sleep by antagonizing adenosine receptors [7,8]. The link between sleep and energy homeostasis is suggested by recent epidemiologic studies, the results of which point to insufficient sleep as a risk factor for future weight gain [9,10,11,12,13,14]. Furthermore, sleep and energy metabolism are closely linked through “multi-tasking” molecules such as orexin, NPY, leptin, insulin, serotonin, and IL6 [15,16]. Since changes in energy metabolism was expected after caffeine/oolong tea consumption in the present study, the crosstalk between regulatory mechanisms of energy metabolism and sleep warrants our experimental design to assess energy metabolism and sleep at the same time.

The effects of caffeine and tea should therefore be evaluated from two perspectives: their effects on energy metabolism and their effects on sleep. The objective of the present study is to assess the effects of subacute ingestion of oolong tea and caffeine on energy metabolism and sleep. To evaluate the effects on the time course of energy metabolism over 24 h, we used whole room indirect calorimetry with an improved time resolution [17]. Sleep was simultaneously monitored by polysomnography during the indirect calorimetry.

## 2. Materials and Methods

### 2.1. Subjects

Twelve subjects were recruited to join this study. All subjects were healthy males, aged 20–56 years of age. Exclusion criteria were body mass index (BMI) < 18.5 kg/m^2^, BMI ≥ 2 5.0 kg/m^2^, food allergies, smoking, chronic diseases, regular use of medications and dietary supplements, shift workers or trans-meridian travel within 1 month, consumer of >300 mg caffeine daily, self-reported sleep problems (Japanese version of the Pittsburgh Sleep Quality Index score > 5), and extreme chronotype (30 < Japanese version of Morningness-Eveningness Questionnaire score < 70).

The study was approved by the Local Ethics Committee of the University of Tsukuba, and conducted in accordance with the ethical principles set forth in the Declaration of Helsinki. We informed the subjects about the experiments and their associated risks, and all subjects provided written informed consent to participate. The trial was registered with the University Hospital Medical Information Network (at https://upload.umin.ac.jp/cgi-open-bin/icdr_e/index.cgi; Registration No. UMIN000035313).

### 2.2. Experimental Beverages

Three types of experimental beverages were prepared: (1) oolong tea containing 51.8 mg caffeine and 48.5 mg catechins, (2) 51.8 mg caffeine, and (3) a placebo. All the experimental beverages were packed in 350-mL cans and supplied from Suntory Beverage and Food Ltd. (Tokyo, Japan). The concentrations of polyphenols and caffeine were analyzed by high-performance liquid chromatography by directly injecting the oolong tea as previously reported [18] (Table 1).

### 2.3. Protocol

The present study was a placebo-controlled, double-blind, cross-over intervention study. During each 14-day session, the subjects consumed two cans of test beverage (350 mL/can; 1 can at breakfast and 1 can at lunch): oolong tea, caffeine, or placebo. During this period, subjects were not allowed to drink other beverages containing caffeine (e.g., coffee and tea) or alcohol. The washout period between the sessions was at least 14 days. During the study, the subjects maintained their usual activities and dietary habits, which were confirmed by activity recordings and a daily diary. All the subjects had one adaptation night in the lab to acclimate to the sensors and experiment room (Figure 1).

Two days before the indirect calorimetry (days 12–13 of each session), experimental meals were provided for breakfast and dinner, but the lunch menu was not controlled. On day 13 of the session, the subjects reported to the laboratory after dinner. The subjects were attached with sensors to monitor their heart rate and swallowed a core body temperature sensor, and then entered the whole room indirect calorimeter, where they were asked to maintain a sitting position while awake and to sleep for 8 h from 23:00 to 7:00. During the indirect calorimetry, the subjects were allowed to use a computer or smartphone or to read a book, but exercise was not allowed. On day 14 of the session, experimental meals were provided as breakfast at 8:00, lunch at 12:00, and dinner at 18:00. The experimental meals were designed to achieve the individual’s energy balance over 24 h during indirect calorimetry. The basal metabolic rate was calculated according to the estimated energy requirements for Japanese individuals [19], and the physical activity factor was assumed to be 1.75 (2432 ± 83 kcal/day) and 1.25 (1737 ± 59 kcal/day) for the pre-calorimetry period (days 12–13) and during calorimetry (day 14), respectively. Experimental meals comprised a total energy intake of 15% protein, 25% fat, and 60% carbohydrate. The contributions of breakfast, lunch, and dinner were 33.3%, 33.3%, and 33.4%, respectively, to the total 24-h intake. Indirect calorimetry was interrupted for 1 h from 21:00 of day 14 of the session to prepare for the polysomnographic measurement. Subjects reentered the metabolic chamber and remained there until 7:00 on day 15. Urine samples were collected during the indirect calorimetry from 7:00 of day 2 to the end of the measurement, and urinary excretion of nitrogen was used to estimate the energy expenditure and substrate oxidation as described below (Figure 1). Body composition was measured by the bioimpedance method (BC-118E, TANITA Co., Tokyo, Japan) before the subjects entered the indirect calorimeter.

### 2.4. Energy Metabolism Measurements

Indirect calorimetry was performed with a room-size metabolic chamber at University of Tsukuba (Fuji Medical Science Co., Ltd., Chiba, Japan) as described previously [20,21]. The metabolic chamber measures 2.00 m × 3.45 m × 2.10 m and has an internal volume of 14.49 m^3^. The chamber is furnished with an adjustable hospital bed, desk, chair, wash basin, and toilet. Meals were provided through a pass-through box. The air flow in the chamber was ventilated at a rate of 80 L/min. The temperature and relative humidity of the chamber were maintained at 25.0 ± 0.5 °C and 55.0 ± 3.0%, respectively. The concentrations of oxygen (O_2_) and carbon dioxide (CO_2_) in the chamber were measured by online process mass spectrometry (VG Prima B, Thermo Electron Co., Winsford, UK). The precision of the mass spectrometry, defined as the standard deviation for continuous measurement of the calibration gas mixture (O_2_ 15%, CO_2_ 5%), was <0.002% for O_2_ and CO_2_.

The O_2_ consumption ($\dot{V}$
O_2_) and CO_2_ production ($\dot{V}$
CO_2_) rates were calculated each minute using an algorithm for improved transient response [17]. Urinary nitrogen was measured using the Kjeldahl method. Energy expenditure and macronutrient oxidation were calculated from the $\dot{V}$
O_2_, $\dot{V}$
CO_2_, and urinary nitrogen excretion. The rate of urinary nitrogen excretion, an index of protein catabolism, was assumed to be constant during the calorimetry [22].

### 2.5. Activity and Sleep Recording

Activity during the intervention period (from day 1 to day 12) was monitored using MTN-220 (Kissei Comtec, Nagano, Japan) [23]. Sleep was recorded by polysomnography, PSG-1100 (Nihon Kohden, Tokyo, Japan). Stage N1, stage N2, stage N3, stage R, and stage W were recorded over 30 s according to the standard criteria [20,21,24,25].

### 2.6. Thermometry

Core body temperatures were continuously recorded by an ingestible core body temperature sensor (CorTemp, HQ Inc, Palmetto, FL, USA) that was 23 × 10.25 mm in size and weighed 2.75 g. The sensor’s signal passed through the body to a recorder that was worn by the subjects around the stomach area. The sensor was accurate to ±0.01 °C and calibrated using hot water before each use [24,26].

### 2.7. Autonomic Nervous System Activity

Autonomic nervous system activity was evaluated from power spectrum analysis of the R-R intervals of the electrocardiogram (LX-3230, Fukuda Denshi Co., Ltd., Tokyo, Japan). The power spectrum of the heart rate variability was estimated using the maximum entropy method. The spectral measures were computed as amplitudes (i.e., areas under the power spectrum) and are presented in square milliseconds (ms^2^). Parasympathetic and sympathetic nervous system activities were estimated as high frequency (HF; 0.15–0.4 Hz) and as the power ratio of the low frequency (LF; 0.04–0.15 Hz) to high frequency (LF/HF), respectively [21,27].

### 2.8. Statistical Analysis

All data are shown as mean ± SE. The effect of the test drink on the time course of energy metabolism, body temperature, heart rate, and autonomic nervous system activity adjusted for the time was analyzed using linear mixed-models analysis of variance (ANOVA) with repeated measures and Bonferroni’s correction for multiple comparisons. As physiologic functions are quite different during wake and sleep, additional statistical analysis was performed separately when awake (7:00–23:00) and asleep (23:00–7:00). Differences in sleep architecture and urinary nitrogen among the three experimental conditions were analyzed by one-way ANOVA. Correlations were assessed using the Pearson correlation coefficient. Statistical analyses were performed using SPSS statistics software (Version 26.0; IBM Corporation) for Apple.

## 3. Results

### 3.1. Physical Characteristics of the Subjects

The characteristics of the subjects are shown in Table 2. Body weight, BMI, and body fat percentage did not change significantly during the entire study. The Morningness-Eveningness Questionnaire score ranged from 44 to 66, suggesting that all of the subjects had a non-extreme chronotype. Body weight and composition, and physical activity was not significantly different among the three trials.

### 3.2. Energy Metabolism

Energy expenditure among the three trials was similar during wake, sleep, or 23 h of indirect calorimetry (Table 3, Figure 2). Compared with the placebo trial, the RQ was lower in the caffeine and oolong tea trials during wake, sleep, or 23 h of indirect calorimetry. During sleep, the RQ was significantly lower in the oolong tea trial than in the caffeine trial. The difference in the hourly RQ between the oolong tea and placebo trials, i.e., effect of oolong tea ingestion on RQ, significantly correlated with the hourly RQ of the placebo trial (*r*^2^ = 0.448, *p* < 0.0001). On the other hand, the effect of caffeine on the RQ was not correlated with the RQ of placebo trial (*r*^2^ = 0.012, *p* = 0.62; Figure 3). Compared with the placebo trial, ingestion of caffeine and oolong tea increased fat oxidation and decreased carbohydrate oxidation during wake, sleep, or 23 h of indirect calorimetry. Urinary nitrogen excretion as an index of protein catabolism was similar among the trials (9.1 ± 0.76, 8.0 ± 0.93, and 8.6 ± 0.67 g/day for placebo, caffeine, and oolong tea trial, respectively, *p* = 0.612).

### 3.3. Core Body Temperature, Autonomic Nervous System Activity, and Sleep Architecture

Compared with the placebo trial, the core body temperature during wake and over 24 h was higher in the caffeine and oolong tea trials, but the difference among the three trials during sleep was not significant (Table 3, Figure 4). Heart rate in the caffeine trial was significantly lower than that in the placebo trial during wake, sleep, and for 24 h, and significantly lower than that in the oolong tea trial during sleep and for 24 h. Sympathetic nervous system activity (LF/HF) over 24 h was significantly lower in the caffeine trial than in the placebo trial. Parasympathetic nervous system activity (HF) during wake, sleep, and over 24 h in the caffeine trial was significantly higher than that in the placebo trial, and significantly higher than that in the oolong tea trial during sleep and over 24 h. Sleep architecture was comparable among the three trials, and there was no statistically significant difference in the sleep parameters (Table 4).

## 4. Discussion

Caffeine ingestion (>100 mg) has an acute effect to increase the energy expenditure [2,3,5,28,29,30,31,32,33,34]. One of the main findings of the present study is that subacute ingestion of caffeine (103.6 mg) did not increase the energy expenditure, suggesting that near complete tolerance to caffeine, in terms of stimulating energy expenditure, was acquired over 2 weeks of caffeine ingestion. Consistent with this notion was the lack of a stimulating effect of subacute ingestion of caffeine on the heart rate and sympathetic nervous system activity in the present study.

Caffeine attenuates the buildup of the homeostatic sleep propensity and delays sleep onset by blocking cerebral adenosine receptors [8,35]. Tolerance to the effects of caffeine on sleep latency, however, develop over 4 days of caffeine ingestion [36]. In the present study, sleep onset latency was not prolonged by subacute ingestion of caffeine. Thus, the effects of tolerance to caffeine on energy expenditure, heart rate, and sleep latency developed over the 2 weeks of caffeine ingestion. On the other hand, subacute ingestion of caffeine alone increased fat oxidation compared to the placebo trial, which is in contrast with the acute effects of caffeine ingestion on 24-h energy metabolism—that is, an increase in energy expenditure without affecting RQ [2,3,4] or no effect on energy expenditure and RQ [5]. Thus, the effect of caffeine ingestion on energy expenditure and fat oxidation was dissociable, but the mechanisms underlying the distinct effect of caffeine on energy expenditure and fat oxidation remain to be clarified. Of note, mean daily caffeine intake of adult male Japanese is 268 ± 176 (mean ± SD) mg, 6% of which is consumed as oolong tea [37]. Dose of caffeine in the present study, 103.6 mg, was within a range of daily caffeine intake.

A positive effect of oolong tea consumption on energy expenditure was observed in two previous studies. One study reported that acute ingestion of oolong tea containing 77 mg caffeine, 81 mg epigallocatechin gallate, and 68 mg polymerized polyphenols increased energy expenditure following 2 h of tea consumption, but did not increase fat oxidation [38]. Another study reported that oolong tea consumption (270 mg/day caffeine, 264 mg/day polymerized polyphenols) increased 24-h energy expenditure and fat oxidation on day 3 of consumption [4]. In the present study, 2 weeks of oolong tea consumption increased fat oxidation, but did not increase the accumulated energy expenditure over 24 h, during sleep or during wake. Together, these findings indicate that over the 2 weeks of oolong tea consumption, the acute effects of oolong tea to increase energy expenditure were diminished but the stimulating effects on fat oxidation continued to be manifested. Experiments monitoring the adaptation process to chronic oolong tea consumption remain to be performed.

Considering half-life of caffeine ~5.7 h [39], effects of oolong tea and caffeine consumption to decrease RQ and increase fat oxidation during sleep require cautious interpretation. The decrease in the RQ during the sleep by oolong tea consumption was significantly greater than that by caffeine ingestion. Visual inspection of the time course of the RQ suggested that the effect of oolong tea to decrease the RQ became clearer when the RQ in the placebo trial was lower; after overnight fasting before breakfast, in the late afternoon, and during sleep, but not immediately after oolong tea consumption at breakfast and lunch. These findings may reflect the slow pharmacokinetics of the substances responsible for the decreased RQ. Alternatively, the susceptibility of energy metabolism to the ingredients of the tea is enhanced during sleep, when fat oxidation is upregulated in a post-absorptive state. On the other hand, the effects of caffeine on the RQ did not correlate significantly with the inherent state of energy metabolism, i.e., the RQ in the placebo trial (Figure 3). The findings of the present study suggest that the effects of unidentified ingredients in oolong tea to decrease the RQ were suppressed during wake. A plausible explanation for our findings is that the strong effects of a meal and subsequent insulin secretion masked the effect of oolong tea to decrease the RQ and stimulate fat oxidation.

The effects of oolong tea consumption to increase fat oxidation during sleep may have clinical relevance for controlling body weight and glucose metabolism. Several lines of evidence suggest that an individual’s capacity to oxidize dietary fat is a metabolic determinant of future weight gain [40]. Mynatt et al. selected metabolically flexible and inflexible subjects based on the difference between 24-h and sleep RQ, and subsequent analysis of global skeletal muscle gene expression suggested a role of the RNA-binding protein HuR as a regulator of metabolic flexibilities in skeletal muscle metabolism [41]. An elevated RQ during sleep may be a phenotype detected at the earliest stage in the pathogenesis of metabolic inflexibility and insulin resistance. The findings of the present study suggest that this phenotype may be corrected by interventions such as oolong tea consumption.

The effects of caffeine and oolong tea to increase the core body temperature were observed only during wake without any increase in energy expenditure, i.e., heat production. It is possible that an increase in the core body temperature by caffeine and oolong tea ingestion is due to reduced heat dissipation. The effects of caffeine and oolong tea on core body temperature are masked when peripheral heat dissipation is upregulated during sleep [42]. The decreased heart rate and upregulated parasympathetic nervous activity by subacute caffeine consumption was not anticipated, and the mechanisms underlying these effects remain to be clarified.

This study has several limitations. First, abstinence from caffeine intake except the experimental beverage was imposed during the 14 days of intervention, but control for energy intake during the experimental 14 days was not perfect (Figure 1). Second, the amount of caffeine consumed in the oolong tea and caffeine trials was matched in the present study to delineate the effective compound in the tea other than caffeine. This approach was used previously to assess the effect of green tea on the energy metabolism [4,5]. In the present protocol, however, the caffeine in the caffeinated water and the tea exerted strong effects on energy metabolism, and analysis of the small difference between the two experimental conditions is vulnerable to experimental error, i.e., the so-called ill-posed question. To further dissect the effective component in the oolong tea, experiments with decaffeinated oolong tea or tablets containing a specific catechin or polymerized polyphenol are warranted. The acute effects of epigallocatechin-3-gallate on energy expenditure and the RQ were assessed using a commercially available product, but no statistically significant effect on energy metabolism was detected [43].

## 5. Conclusions

The effects of caffeine and oolong tea ingestion for 14 days on energy metabolism were different from the findings of previous studies assessing their acute effects (2–5), which are particularly important when considering the habitual nature of tea consumption. Two weeks was not long enough to assess the effects of the intervention on body weight and body composition, and whether caffeine or oolong tea consumption actually reduces body fat remains to be evaluated.

## Figures and Tables

**Figure 1 nutrients-12-03671-f001:**
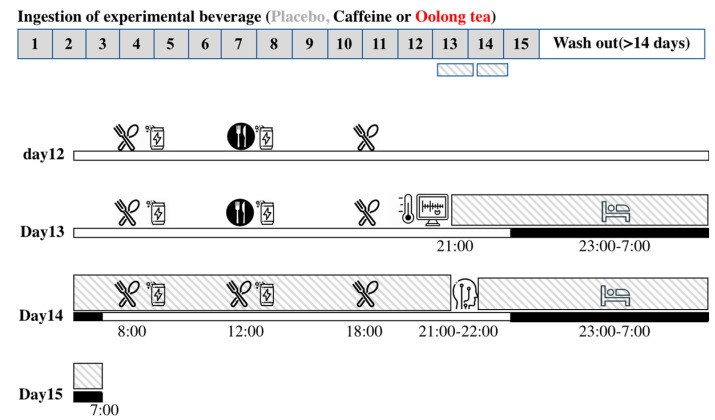
Experimental protocol. Schematic overview of the study protocol (upper panel) and time schedule during indirect calorimetry (lower panel). During each 14-day session, subjects consumed two cans of the test beverage (one at breakfast and one at lunch): oolong tea, caffeine, or placebo, and physical activity was monitored on day 1–12. The washout period between sessions was ≥14 days. Two days before the indirect calorimetry (day 12 to 13 of each session), experimental meals were provided for breakfast and dinner (
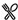
), but the lunch menu was not controlled (
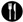
). On day 13 of the session, the subjects wore sensors for heart rate, swallowed a core body temperature sensor, and then entered the whole room indirect calorimeter (
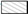
), where they slept for 8 h from 23:00 to 7:00 (
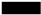
). On day 14 of the session, experimental meals were provided as breakfast at 8:00, lunch at 12:00, and dinner at 18:00. Indirect calorimetry was interrupted from 21:00 of the 14th day of the session for preparing polysomnographic measurement.

**Figure 2 nutrients-12-03671-f002:**
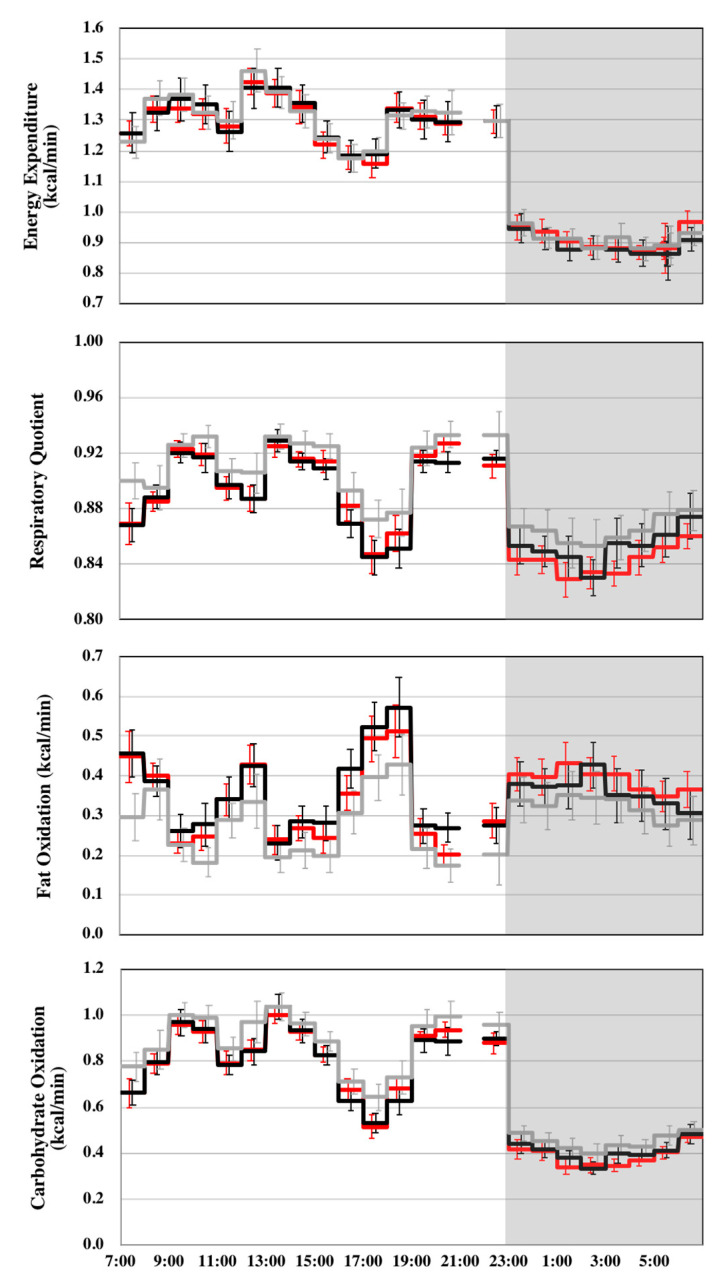
Hourly mean energy metabolism. Values are mean ± SE for placebo (**-**), caffeine (**-**), and oolong tea trial (**-**). Indirect calorimetry was interrupted for 1 h from 21:00 of day 14 of the session for preparation of the polysomnographic measurement.

**Figure 3 nutrients-12-03671-f003:**
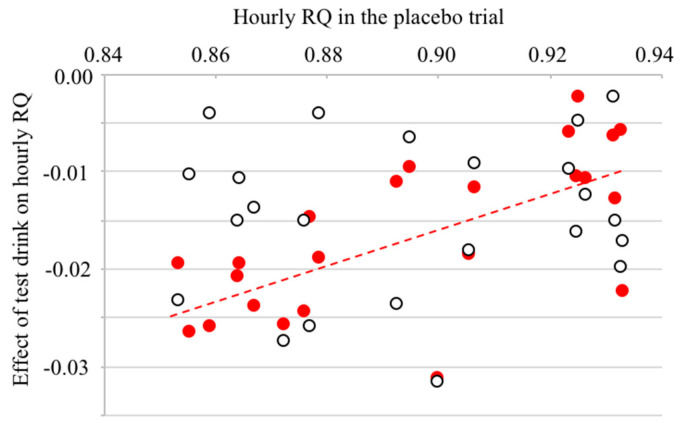
Relation between hourly mean respiratory quotient (RQ) in a placebo trial and effect of the test drink on RQ. Difference in the hourly mean RQ between placebo and oolong tea trial was significantly correlated with hourly mean RQ in the placebo trial (●, *r*^2^ = 0.448, *p* < 0.001). The effects of caffeine on the hourly mean RQ and hourly mean RQ in the placebo trial were not significantly correlated (○, *r*^2^ = 0.012, *p* = 0.62).

**Figure 4 nutrients-12-03671-f004:**
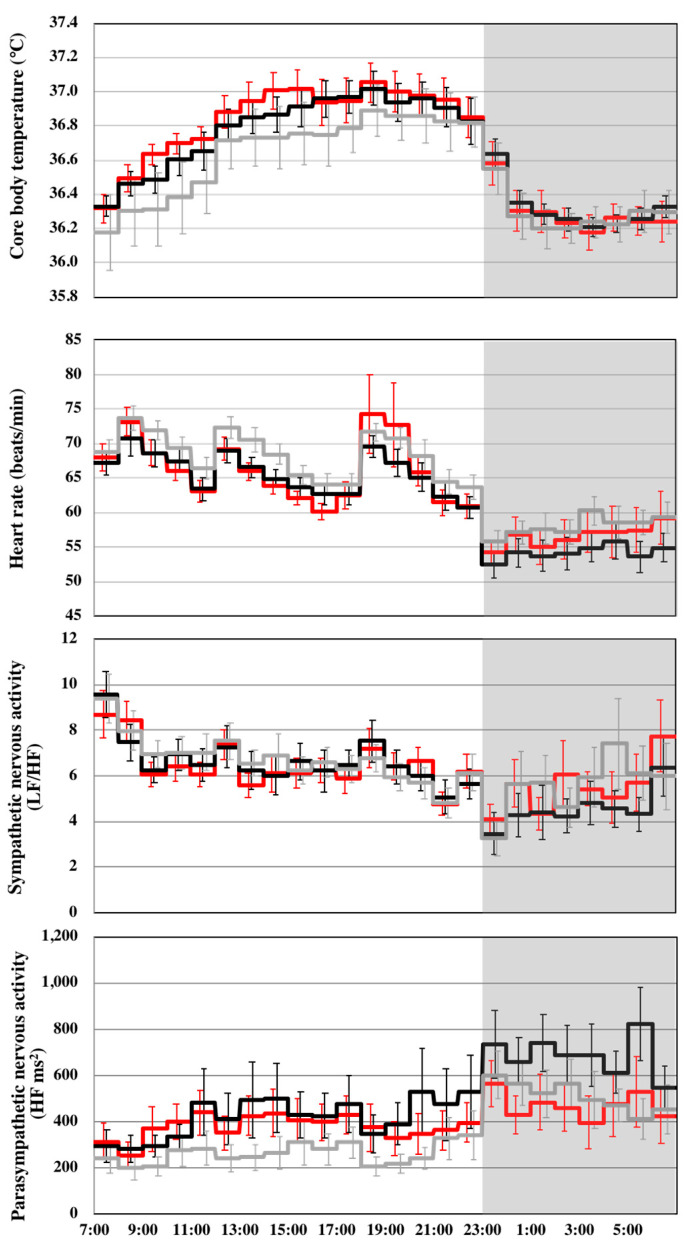
Core body temperature, heart rate, and autonomic nervous system activity. Values are mean ± SE for placebo (**-**), caffeine (**-**), and oolong tea trial (**-**).

**Table 1 nutrients-12-03671-t001:** Composition of oolong tea consumed by the subjects *.

	Placebo	Caffeine	Oolong Tea
Caffeine	0.0	51.8	51.8
Gallic acid	0.0	0.0	10.7
Catechins			
Cathechin	0.0	0.0	3.4
Gallocatechin	0.0	0.0	12.3
Catechin gallate	0.0	0.0	2.9
Gallocatechin gallate	0.0	0.0	10.5
Epicatechin	0.0	0.0	1.8
Epigallocatechin	0.0	0.0	5.2
Epicatechin gallate	0.0	0.0	2.4
Epigallocatechin gallate	0.0	0.0	10.0
Other polyphenols including			
polymerized polyphenols	0.0	0.0	62.3

* Concentration of the ingredients is expressed as mg/350 mL of the test drink.

**Table 2 nutrients-12-03671-t002:** Characteristics of the subjects.

	Placebo	Caffeine	Oolong Tea
Age (years)	37.1 ± 4.3		
Height (cm)	168.2 ± 1.7		
PSQI-J	1.4 ± 0.5		
MEQ-J	53.0 ± 2.5		
Weight (kg)	62.0 ± 1.5	61.6 ± 1.6	61.6 ± 1.5
BMI (kg/m^2^)	21.9 ± 0.6	21.8 ± 0.5	21.8± 0.5
Body fat (%)	18.5 ± 1.0	17.8± 1.0	17.5± 1.1
Body water (%)	54.9 ± 1.1	55.5 ± 1.1	56.5 ± 1.4
Physical activity (kcal/day)	1843 ± 63	1850 ± 49	1817 ± 50

Values are means ± SE. Weight, BMI, and body fat % were measured before indirect calorimetry of each trial. Physical activity was measured from day 1 to day 12 of intervention period. PSQI-J: Japanese version of Pittsburgh Sleep Quality Index score. MEQ-J: Japanese version of Morningness-Eveningness Questionnaire score.

**Table 3 nutrients-12-03671-t003:** Effect of subacute ingestion of caffeine and oolong tea on energy metabolism, core body temperature, heart rate, and autonomic nervous system activity.

	Placebo	Caffeine	Oolong	Df ^#^	F ^#^	*p*-Value ^#^
**24 h**						
Energy expenditure (kcal/min)	1.17 ± 0.26	1.16 ± 0.27	1.16 ± 0.24	748.0	1.175	*p* = 0.309
RQ	0.90 ± 0.05	0.88 ± 0.05 **	0.88 ± 0.047 **	748.0	24.792	*p* < 0.001
Fat oxidation (kcal/min)	0.29 ± 0.20	0.36 ± 0.20 **	0.35 ± 0.17 **	748.0	26.484	*p* < 0.001
Carbohydrate oxidation (kcal/min)	0.74 ± 0.29	0.68 ± 0.27 **	0.67 ± 0.27 **	748.0	18.437	*p* < 0.001
Core body temperature (°C)	36.5 ± 0.5	36.6 ± 0.4 **	36.7 ± 0.5 **	721.3	11.048	*p* < 0.001
Heart rate (beats/min)	65 ± 8	62 ± 8 **	64 ± 10 ^††^	599.2	10.217	*p* < 0.001
LF/HF	6.85 ± 3.03	5.88 ± 2.86 *	6.31 ± 2.79	599.5	3.396	*p* = 0.34
HF (ms^2^)	315 ± 259	502 ± 394 **	393 ± 296 ^††^	598.4	19.312	*p* < 0.001
**Wake time**						
Energy expenditure (kcal/min)	1.31 ± 0.20	1.30 ± 0.21	1.30 ± 0.17	484.0	0.903	*p* = 0.406
RQ	0.91 ± 0.05	0.90 ± 0.04 **	0.90 ± 0.04 **	484.0	16.950	*p* < 0.001
Fat oxidation (kcal/min)	0.27 ± 0.20	0.35 ± 0.20 **	0.33 ± 0.18 **	484.0	21.930	*p* < 0.001
Carbohydrate oxidation (kcal/min)	0.89 ± 0.24	0.82 ± 0.22 **	0.82 ± 0.20 **	484.0	12.820	*p* < 0.001
Core body temperature (°C)	36.6 ± 0.6	36.8 ± 0.4 **	36.8 ± 0.4 **	481.0	12.329	*p* < 0.001
Heart rate (beats/min)	68 ± 6	66 ± 6 **	67 ± 9	395.6	5.083	*p* = 0.007
LF/HF	7.21 ± 2.41	6.49 ± 2.52	6.58 ± 2.37	394.5	3.239	*p* = 0.040
HF (ms^2^)	252 ± 201	427 ± 372 **	366 ± 266 **	393.7	21.153	*p* < 0.001
**Sleep**						
Energy expenditure (kcal/min)	0.91 ± 0.13	0.89 ± 0.14	0.91 ± 0.11	253.0	2.312	*p* = 0.101
RQ	0.86 ± 0.05	0.85 ± 0.05 **	0.84 ± 0.04 **^,†^	253.0	15.420	*p* < 0.001
Fat oxidation (kcal/min)	0.32 ± 0.20	0.36 ± 0.20 *	0.39 ± 0.15 **	253.0	12.391	*p* < 0.001
Carbohydrate oxidation (kcal/min)	0.45 ± 0.13	0.41 ± 0.13 **	0.39 ± 0.12 **	253.0	11.093	*p* < 0.001
Core body temperature (°C)	36.3 ± 0.4	36.3 ± 0.3	36.3 ± 0.4	230.8	0.413	*p* = 0.662
Heart rate (beats/min)	58 ± 6	54 ± 7 **	56 ± 9 ^††^	194.1	9.963	*p* < 0.001
LF/HF	5.59 ± 3.95	4.58 ± 3.01	5.28 ± 3.44	196.3	3.664	*p* = 0.027
HF (ms^2^)	511 ± 290	684 ± 380 *	472 ± 346 ^††^	200.1	8.800	*p* < 0.001

RQ: Respiratory Quotient. Values are means ± SE. ^#^ Denominator degrees of freedom (Df) and F for linear mixed-models ANOVA, and *p*-value for main treatment effect of beverage. *, ** significantly different from placebo trial (*p* < 0.05, *p* < 0.01). ^†^, ^††^ significant difference between caffeine and oolong tea trial (*p* < 0.05, *p* < 0.01).

**Table 4 nutrients-12-03671-t004:** Sleep architecture.

	Placebo	Caffeine	Oolong Tea
Time in bed (min)	480.0	480.0	480.0
Sleep period (min)	468.6 ± 3.8	464.6 ± 5.8	460.0 ± 6.8
Total sleep time (min)	419.4 ± 11.4	424.9 ± 11.1	411.6 ± 14.9
Sleep efficiency (%)	87.4 ± 2.4	88.6 ± 2.3	85.7 ± 3.1
Sleep latency (min)	11.0 ± 3.8	6.1 ± 1.7	12.5 ± 6.2
Wake after sleep onset (min)	49.6 ± 12.1	48.5 ± 11.4	56.0 ± 15.9
REM latency * (min)	92.4 ± 7.6	84.7 ± 11.5	107.8 ± 15.5
N1 (min)	73 ± 9.8	68.8 ± 10.6	56.7 ± 7.3
N2 (min)	199.3 ± 9.7	216.0 ± 12.1	200.1 ± 10.0
N3 (min)	61.3 ± 13.9	55.5 ± 14.6	69.5 ± 15.7
R (min)	85.8 ± 5.4	84.6 ± 3.2	85.4 ± 7.6

Values are means ± SE. * REM latency from sleep onset epoch to first REM epoch.

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
