# Peer review of "Subacute Ingestion of Caffeine and Oolong Tea Increases Fat Oxidation without Affecting Energy Expenditure and Sleep Architecture: A Randomized, Placebo-Controlled, Double-Blinded Cross-Over Trial"

_nutrients, 2020, doi:10.3390/nu12123671_

Round 1

Reviewer 1 Report

Thank you for allowing me to review this submitted manuscript. The article was well written and provided a decent background for the methodology. The figures and tables added to the text and were clearly labeled and provided good information to stand alone. The methodology was inclusive but could have been more clear on a few items as noted below.

One area of concern is the over-exaggeration of the results. Clinically, will it make much of a difference if RQ changes from 0.90 to 0.88? That is still in high CHO oxidation and does not increase the fat oxidation by that much. The over-emphasise on sleep is a concern as well since average half-life of caffiene is 5 hours and it is approx. 7 hours from their last ingestion to sleep. So it should not affect the adenosine receptors at that point. I also would like to see more control of diet and physical activity during tehe 14 days. These highly impact EE and sleep. Please see my comments below.

Line 14. Might be beneficial to be specific with what ingredients. You also state that it does affect when previously you state that the evidence is inconclusive.

Line 20: Yes I agree, however, the half-life of caffeine is short so your study ending caffeine/tea consumption at lunch should not affect sleep.

Line 22: Lack of sleep has more effects than just weight gain. Why focus on just that when that is not the main objective of your study.

Line 23: Lack of sleep has more effects than just on insulin and orexin. I would suggest sticking with the topic. The authors introduce some major concepts that can lead down a rabbit hole without providing enough context or description to support the roles.

Since it seems like catechins were a primary objective in the study then I would suggest developing the evidence for this polyphenol to support the use of them.

Line 34: Is this all food allergies?

Line 112: Did you control for hydration status before the BIA measure?

How did you control for energy intake during the experimental 14 days? They were not allowed to drink anything with caffeine in it what about foods that contain caffeine of catechins?

There is no mention of outside calories or macros?. Do you have any data for that?

Did you control for physical activity? We know that an increase in exercise before total room calorimetry can change RQ and EE.

Line 309: How can you claim this if you did not test the acute single intake of caffeine on EE with this test group.

Line 314: Again, this is difficult to determine since they stopped the ingestion 7 hours before sleep. According to Richter 1992 and Busto 1989 the mean have life in plasma if healthy individuals are about 5 hours.

Line 376: When you say their effects from another study, you can generalize but you can't compare since these are different people in a different environment.

Author Response

Thank you for allowing me to review this submitted manuscript. The article was well written and provided a decent background for the methodology. The figures and tables added to the text and were clearly labeled and provided good information to stand alone. The methodology was inclusive but could have been more clear on a few items as noted below. Response: We appreciate your comments, which helped to clarify the results and discussion of our manuscript.

One area of concern is the over-exaggeration of the results. Clinically, will it make much of a difference if RQ changes from 0.90 to 0.88? That is still in high CHO oxidation and does not increase the fat oxidation by that much. The over-emphasise on sleep is a concern as well since average half-life of caffiene is 5 hours and it is approx. 7 hours from their last ingestion to sleep. So it should not affect the adenosine receptors at that point. I also would like to see more control of diet and physical activity during the 14 days. These highly impact EE and sleep. Please see my comments below. Response: Effects of caffeine/oolong tea consumption on RQ is small but statistically significant. In terms of fat oxidation, it was increased in caffeine/oolong tea consumption trial by 21-24%; ~90 kcal/24h (Table 3). However, there is no guarantee that the acute effects of caffeine/oolong tea consumption on fat metabolism continue to be manifested when caffeine/oolong tea consumption is prolonged. As a limitation of this study, the last sentence was put in the original manuscript [line 356-358]. Please see our response to your comment on “over-emphasis on sleep”, control of diet and physical activity.

Line 14. Might be beneficial to be specific with what ingredients. You also state that it does affect when previously you state that the evidence is inconclusive. Response: We modified the sentence as “Thus, unidentified ingredient(s) of tea other than caffeine affect…” [line 14]. Discussion in one study using green tea extract focused on the role of (-)-epigallocatechin gallate, but that study didn’t prove the role of this polyphenol, since green tea extract contained several polyphenols and unidentified ingredients (reference5). So far, the study using semi-purified epigallocatechin-3-gallate observed no effects of this compound on energy metabolism [line 347-351] (reference 43). We completely agree with the reviewer’s comment that developing the evidence for (particular) polyphenol is necessary, but it remains to be done. Limitation of previous and present study was described [line 339-351].

Line 20: Yes I agree, however, the half-life of caffeine is short so your study ending caffeine/tea consumption at lunch should not affect sleep Response: Considering half-life of caffeine ~5h, the possibility that caffeine/tea consumption at lunch affect sleep is low but can’t be excluded, i.e., roughly one quarter of caffeine remains in the body at bedtime. Furthermore, the effects of caffeine/tea consumption on substrate oxidation, heart rate and autonomic nervous system activity during sleep suggest residual effects of caffeine/tea ingestion during sleep (Table 3). In the context of the residual effect of caffeine/tea on sleeping energy metabolism, the issue raised by the referee was discussed citing a new reference on half-life of caffeine (reference 39)[line 308-309].

Line 22: Lack of sleep has more effects than just weight gain. Why focus on just that when that is not the main objective of your study. Response: As reviewed in reference 15, the concurrent background level of metabolic activity may control state of vigilance, promoting wakefulness (and hunger) when it is low, or sleep (and satiety) when it is high. As changes in energy metabolism was expected after caffeine/oolong tea consumption in the present study, a crosstalk between regulatory mechanisms of energy metabolism and sleep warrants our experimental design to assess energy metabolism and sleep at the same time. Although caffeine/oolong tea consumption didn’t affect sleep, it was important to make sure that sleep is not disturbed by the dietary intervention. We appreciate your comment, and sentence was added to explain our point of view [line 22-26].

Line 23: Lack of sleep has more effects than just on insulin and orexin. I would suggest sticking with the topic. The authors introduce some major concepts that can lead down a rabbit hole without providing enough context or description to support the roles. Response: It is correct that not only insulin and orexin, but leptin, ghrelin, IL6 and serotonin also play a role in regulation of energy metabolism and sleep (reference 15). Changes in energy metabolism was expected after caffeine/oolong tea consumption in the present study. The crosstalk between regulatory mechanisms of energy metabolism and sleep warrants our experimental design to assess energy metabolism and sleep at the same time. Sentence was added to explain our point of view [line 22-26].

Since it seems like catechins were a primary objective in the study then I would suggest developing the evidence for this polyphenol to support the use of them. Response: Discussion in one study using green tea extract focused on the role of (-)-epigallocatechin gallate. However, that previous study didn’t prove the role of this polyphenol, since green tea extract contained several polyphenols and unidentified ingredients (reference 5). So far, the study using semi-purified epigallocatechin-3-gallate observed no effects of this compound on energy metabolism (reference 43). The reviewer is right that developing the evidence for polyphenol is necessary, but it remains to be done. Limitation of previous and present study was described in the original manuscript [line 339-351].

Line 34: Is this all food allergies? Response: Yes, we asked for all food allergies [line 37].

Line 112: Did you control for hydration status before the BIA measure?Response: No, hydration status was not controlled, but there was no significant difference in body water among the 3 conditions. Data was added on Table 2.

How did you control for energy intake during the experimental 14 days? They were not allowed to drink anything with caffeine in it what about foods that contain caffeine of catechizes? Response: Comment by the referee was well taken, and we added a sentence as a limitation of the study [line 339-341].

There is no mention of outside calories or macros?. Do you have any data for that? Response: Macronutrient composition of the experimental meal (if this is what the referee asked) was 15% protein, 25% fat, and 60% carbohydrate [line 105-106].

Did you control for physical activity? We know that an increase in exercise before total room calorimetry can change RQ and EE. Response: We asked not to modify daily habit including physical activity during the study. Physical activity before calorimetry was not significantly different among the 3 trials. A sentence was added and data was put on Table 2 [line 84, 133-134].

Line 309: How can you claim this if you did not test the acute single intake of caffeine on EE with this test group. Response: It has been repeatedly shown that caffeine ingestion has an acute effect to increase energy expenditure (reference 2,3,5,28-34). Although many of previous studies used higher dose of caffeine (200 ~500 mg/day) compared to that of the present study (103.6mg/day). However, Astrup et al reported acute effect of 100mg of caffeine to increase energy expenditure (reference 31). The comment by the referee was well taken, and sentences were modified to add dose of caffeine in previous studies [line 278], and slightly modified the first sentence of conclusion [354-355].

Line 314: Again, this is difficult to determine since they stopped the ingestion 7 hours before sleep. According to Richter 1992 and Busto 1989 the mean have life in plasma if healthy individuals are about 5 hours. Response: Considering half-life of caffeine ~5h, the possibility that caffeine/tea consumption at lunch affect sleep is low but can’t be excluded, i.e., roughly one quarter of caffeine remains in the body at bedtime. The sentence was slightly modified to acknowledge referee’s point of view [line 308-309]. Please see also our response to referee’s comments to line 20, 22 and 23.

Line 376: When you say their effects from another study, you can generalize but you can't compare since these are different people in a different environment. Response: Acute effect of caffeine ingestion on energy expenditure and heart rate have been repeatedly demonstrated. But many effects of caffeine diminish when caffeine ingestion continues, known as “caffeine tolerance”. Wording of the sentence was slightly modified to response to your comment [line 354-355].

Reviewer 2 Report

This is an interesting study.  Given the amounts of caffeine ingested per day were relatively low (~100mg) I would like to see some discussion of this in the context of normal consumption levels of both caffeine and oolong tea.

Abstract

  • It would be good to include the amounts of caffeine ingested/day in each treatment

Introduction

Well written and interesting

Materials and Methods

Table 1- move the mg/350ml to legend or include on second line- this will tidy up the table on line 70

Line110-111 include what analyses the urine samples were collected for

Results

Table 3 align p values in column

Line 242 body not vody  also activity not activiy

Discussion

Include some discussion putting the results in the context of normal consumption levels of caffeine and oolong tea

Author Response

This is an interesting study. Given the amounts of caffeine ingested per day were relatively low (~100mg) I would like to see some discussion of this in the context of normal consumption levels of both caffeine and oolong tea. Response: Mean caffeine intake of adult male Japanese is 268.3 mg/d for, 6% of which is consumed as oolong tea (reference 37). Sentences were added [line 294-296].

Abstract. It would be good to include the amounts of caffeine ingested/day in each treatment. Response: The information on the ingredients of both drinks was added in the abstract.

Introduction Well written and interesting.Response:  Thank you.

Materials and Methods. Table 1- move the mg/350ml to legend or include on second line- this will tidy up the table on line 70.Response: Table 1 was modified as suggested.

Line110-111 include what analyses the urine samples were collected for. Response: Urine was collected to assess 24 h urinary excretion of nitrogen, which is required for calculation of indirect calorimetry [line 127-130]. An explanation was added [line 109-112].

Results. Table 3 align p values in column. Response: P values was misaligned during the process of conversion to PDF. This time, we made sure of the process.

Line 242 body not vody  also activity not activity. Response: Typographical errors were corrected.

Discussion. Include some discussion putting the results in the context of normal consumption levels of caffeine and oolong tea.Response: Mean caffeine intake of adult male Japanese is 268.3 mg/d for, 6% of which is consumed as oolong tea (reference 37). Sentences were added [line 294-296].

Reviewer 3 Report

In this article, the authors aimes to assess the effects of subacute ingestion of oolong tea and caffeine on energy metabolism and sleep. To evaluate the effects on the time course of energy metabolism over 24 h, they used whole room indirect calorimetry with an improved time resolution. Sleep was simultaneously monitored by polysomnography during the indirect calorimetry.

Because of the incerased consumtion of caffeine and the need of determination of chronic effect of caffeine, this work is pertinent. the methods used are complex and difficult to handle together. The multidisciplinary approach of the methods used is original.

The work was well conducted and the results clearly expressed.

Abstract.

I suggest giving some numerical results (order of magnitude of the difference... significant effect) and not just a literal expression of the results. 

Methods.

Pleace, specify the length of abstinence (withdrawal) from tea and coffee before the study. How did you determine this duration? 

Results.

In the table 2, please give us the habitual caffeine, café and/or tea consumtion 

In supplementary data, please give us a table with the main ANOVA effects.

Please, explain what is the p valu foe ANVA, the main treatment effect? Give us the freedom degrees.

Did you observed a shift ad/or a delay in the circadian rythm? (increased sleep onset?)

Author Response

In this article, the authors aimes to assess the effects of subacute ingestion of oolong tea and caffeine on energy metabolism and sleep. To evaluate the effects on the time course of energy metabolism over 24 h, they used whole room indirect calorimetry with an improved time resolution. Sleep was simultaneously monitored by polysomnography during the indirect calorimetry. Because of the incerased consumtion of caffeine and the need of determination of chronic effect of caffeine, this work is pertinent. the methods used are complex and difficult to handle together. The multidisciplinary approach of the methods used is original. The work was well conducted and the results clearly expressed. Response: Thank you.

Abstract. I suggest giving some numerical results (order of magnitude of the difference... significant effect) and not just a literal expression of the results. Response: We tried our best within the limit of 200 words for abstract.

Methods. Pleace, specify the length of abstinence (withdrawal) from tea and coffee before the study. How did you determine this duration? Response: During the 2 weeks of ingestion of experimental beverage, subjects were instructed not allowed to drink beverages containing caffeine (e.g., coffee and tea). The sentence was slightly modified and moved to clarify length of abstinence from tea and coffee [line 74-75].

Results.In the table 2, please give us the habitual caffeine, café and/or tea consumption. Response: Mean caffeine intake of adult male Japanese was 268.3 mg/d for, 6% of which is consumed as oolong tea (reference 37). We didn’t asses habitual caffeine, café and/or tea consumption.

In supplementary data, please give us a table with the main ANOVA effects. Please, explain what is the p valu foe ANVA, the main treatment effect? Give us the freedom degrees. Response: P values on the table is for the main treatment effect by mixed linear ANOVA. Degree of freedom and F were also added in Table 3 [line 205-230, 232].

Did you observed a shift ad/or a delay in the circadian rythm? (increased sleep onset?) Response: Sleep onset was not significantly affected by caffeine and oolong tea, i.e., sleep latency in was similar among the trials [Table 4, line 268].